# Exploring Methods to Evaluate HPAI Transmission Risk in Iowa During Peak HPAI Incidence, February 2022–December 2023

**DOI:** 10.3390/ijerph22030400

**Published:** 2025-03-10

**Authors:** Christopher Jimenez, Lori A. Hoepner

**Affiliations:** Department of Environmental and Occupational Health Sciences, School of Public Health, SUNY Downstate Health Sciences University, Brooklyn, NY 11203, USA; lori.hoepner@downstate.edu

**Keywords:** zoonoses, animal–human–environment triad, planetary health

## Abstract

Highly pathogenic avian influenza (HPAI), H5N1 strain, began to circulate in the United States on 8 February 2022. The state of Iowa lost the most domestic poultry to HPAI from February 2022–December 2023. This study conducted preliminary evaluations on two environmental risk factors, (inland water surface area, Canada geese abundance) and the availability of the data needed to evaluate them. Higher Canada geese abundance was significantly associated (X^2^ = 4.29, *p* = 0.04) with HPAI negative counties. Farm location data were unavailable, limiting our analysis. Van den Broeck et al.’s framework was used to evaluate the available data. Outcome data from Animal and Plant Health Inspection Service (APHIS) had the highest data quality score (11). Canada geese and inland water surface area are predictors worth evaluating, but poultry farm location data are needed for a comprehensive evaluation.

## 1. Introduction

A historic public health event occurred in the United Sates (US) during the current highly pathogenic avian influenza (HPAI) epizootic, with the first-ever confirmed human HPAI case in the US being reported in the spring of 2022 [1]. HPAI virus cannot infect humans efficiently. However, individuals who work closely with infected poultry and other domestic birds, such as ducks and geese raised in backyards, can contract these viruses [2]. Inland water and Canada geese may play a role in maintaining HPAI incidence because the current consensus is that the US epizootic (HPAI-21) was seeded by wild waterfowl, like Canada geese. This species of bird, like all species of waterfowl, instinctually seeks out inland water [3,4].

The primary objective of this study was the evaluation of two natural environmental factors (inland water, Canada geese abundance) and their association to HPAI risk. Our hypotheses seek to answer the question of whether inland water or Canada geese abundance are significantly associated with HPAI risk. This study posits that common interactions between wild birds at inland water bodies, resulting in the infection of susceptible birds such as Canada geese, increases HPAI risk for susceptible poultry farms. We also evaluated the process to obtain the data used to evaluate those risk factors (Figure 1). If poultry farm location data are available, we evaluate whether the distance of poultry farms from inland water bodies and abundant populations of Cananda geese are associated with higher HPAI risk. If location data are not available, we evaluate whether cumulative inland water surface area and cumulative Canada geese counts per county are associated with a higher risk of an HPAI outbreak.

The Semantic diagram (Figure 1) provides a visual explanation of what this study evaluates: possible natural environmental risk factors and their available data sources. HPAI-21 is the latest example of an event that elucidates the relevance of the One Health approach [5]. The health of humans, animals, and the environment are interconnected, and this epizootic began and continues to be maintained by that interconnectivity. This interconnectivity is visualized on the One Health approach conceptual diagram that incorporates the specific elements of this study (Figure 2) [5]. The diagram visually displays the interconnectivity between inland water, the species of interest (Canada geese), susceptible domestic poultry farms, and susceptible humans working on those farms.

### 1.1. Summary of HPAI-21 Epizootic

HPAI strain H5N1 was first detected among wild birds in Canada in November 2021 (HPAI-21). Wild birds in the US then tested positive for the same strain in January 2022 [3]. The first outbreaks among US domestic poultry chickens, domesticated turkeys, geese, and ducks were reported in February 2022 [6]. By April 2022, the first confirmed human case of HPAI in the US was reported in the state of Colorado [7]. Individuals who work in close quarters with infected poultry, whether the birds are dead or alive, can contract the virus [2]. Direct infection can occur from exposure to saliva, mucous, or fecal matter from infected birds [2]. Individuals who do not wear respiratory and eye protection, when encountering infected birds, are at a higher risk for HPAI infection [2]. H5N1 is the strain of HPAI that has been circulating among wild and domestic birds in the US during this epizootic [1].

The continued incidence and prevalence of HPAI in the US provides justification for evaluating understudied natural environmental factors, such as inland water and Canada geese abundance. Evaluating these factors may provide novel information that stakeholders can use when deciding to reinforce, or change, prevention methods. Data acquisition and data quality evaluation are also necessary because stakeholders need to know if the current publicly available data, along with the standards and methods to access the data, are appropriate.

### 1.2. HPAI Risk to Public Health: Mutation

There is no evidence of human-to-human transmission for H5N1, or any strain of HPAI [1]. However, human-to-human transmission might eventually occur as all strains of HPAI are consistently mutating [8,9]. A novel strain emerging through mutation can occur via two pathways: continued high prevalence of HPAI among wild and domestic birds resulting in “mixing vessel mammals” creating the novel strain, or antiviral flu medication being overused by medical providers treating humans diagnosed with HPAI [10,11].

A high prevalence of the HPAI among wild and domestic birds, as there was during the peak (February 2022–December 2023) of HPAI-21 in Iowa, will result in consistent exposure of susceptible “mixing vessel” mammals, and humans [12,13]. A “mixing vessel” is a mammal that can contract human influenza A viruses and HPAIs simultaneously [2,12]. During co-infections the viruses can swap genes, resulting in a novel strain capable of human-to-human transmission because it contains genes from both human influenza A and HPAI [10]. Mixing vessel mammals can contract human influenza A and HPAI, simultaneously, via direct contact from infected humans that work with them, and through contact with infected organic wild bird material [2,12]. Waterfowl can encounter mixing vessels on farms because most waterfowl, including Canada geese, are attracted to livestock feed grains and the standing water found on farms [3,14].

The current scientific consensus is that pigs, bats, and minks are mammals that can function as mixing vessels. Of those mammals, pigs and minks are considered the mammals to prioritize for HPAI risk reduction [9]. Zhao et al. conducted a 2019 study which reported that influenza databases showed that pigs had significant numbers of human and avian influenza strains present in their sample cells [9]. These results provided evidence that susceptible pigs can become co-infected with multiple influenza A strains. Minks had the most avian and mammal influenza strains isolated in their sample cells. These results provided evidence that they also can be co-infected, with the concern being that while they are not common livestock like pigs, their behaviors might lead to them encountering significant amounts of infectious material from mammals and wild birds [9]. H5N1 has continued to circulate in the US since its 2022–2023 peak. The endemic prevalence of H5N1 has increased exposure and infections among all mammals, including humans, creating conditions for a novel strain to emerge [1].

A common consideration to reduce risk of human-to-human transmission of HPAI is to create a human vaccine to provide protection from H5N1, much like the current vaccine for seasonal human influenza A. However, as the 2024 study from Subedi et al. elucidates, creating a vaccine for avian influenza is challenging [15]. The dissimilarity in the genetic base of commercial vaccines and the dominant strain of the HPAI will result in an ineffective vaccination [15]. To create the most effective HPAI vaccine for humans, it is best for manufactures to know the exact HPAI strain facilitating human-to-human transmission. By the time that occurs, the mutated novel strain is already circulating among the general population and will continue to mutate as it infects more human hosts. Therefore, decreasing HPAI incidence and human exposure now is the more practical upstream strategy. 

Another pathway for HPAI to mutate into a strain capable of human-to-human transmission is through medical providers over-prescribing antiviral drugs for individuals with HPAI symptoms. A 2023 Pakistani study conducted by Ifitkhar et al., concluded that HPAI virus H9N2 mutated under drug pressure [11]. The antiviral drug evaluated in that study, Oseltamivir, is the same drug the Center for Disease Control and Prevention (CDC) currently recommends for use in individuals infected with HPAI in the US [16].

CDC guidelines on treating individuals exposed to, or displaying symptoms of HPAI H5N1, urge medical providers to promptly prescribe Oseltamivir [17]. The guidelines for outpatient care specifically state that the drug should be given “as soon as possible” to individuals confirmed or suspected of H5N1 infection [17]. These guidelines create the perception that rapid prescription of Oseltamivir is acceptable and beneficial. However, the biproduct of this perception is the increased likelihood of H5N1 mutating due to drug pressure. While it is important to treat individuals infected with a possible deadly virus, promoting rapid prescription of Oseltamivir for most cases may generate a new challenge in the form of creating a novel HPAI strain capable of human-to-human transmission.

The best way to reduce the effect that HPAIs can have on public health is upstream intervention. Our study is focused on evaluating understudied natural environmental risk factors for that reason. Providing evidence to help stakeholders determine which HPAI prevention strategies to utilize supports upstream efforts to reduce HPAI incidence and prevalence; thereby limiting human and animal exposure. Reducing HPAI exposure will reduce the risk of infection among vectors, mixing vessel animals, and farm workers. Fewer infections will result with less opportunities for the virus to mutate, and this will also decrease the need for antiviral drugs and vaccines.

### 1.3. Inland Water as a Risk Factor

Three previous Asian studies (two Japanese, one Korean) provided evidence that inland water bodies are places where waterfowl, the reservoirs and vectors of HPAI, commonly congregated [18,19,20]. When infected waterfowl congregate at inland water bodies close to susceptible poultry farms, they are within range to deposit infected organic material on those farms, or to expose and infect susceptible animals that can do the same [18,19,20]. These previous studies elucidate the relevance of this current study; that inland water bodies in Iowa should be prioritized and evaluated as a primary risk factor for HPAI. Iowa is a state rich in inland water, making it is an attractive location for waterfowl [21]. The 2018 study conducted by Shimizu et al. had access to on-site data gathered by Japan’s veterinary authorities [18]. With these data, the authors were able to produce statistically significant results which provided evidence that farms were at a higher risk for HPAI outbreaks the closer they were to inland water [18]. The study also reported that inland water bodies were locations where intermingling of various species created conditions that made it possible for that area to have higher HPAI incidence [18]. Iowa’s location along the Mississippi flyway places it in prime position for high risk comingling to occur, given the high number of inland water bodies, waterfowl and poultry farms found in state [22,23,24]. A second Japanese study from Yamaguchi et al., in 2024, yielded similar results to the Shimizu et al. study [19]. Results from this study indicated that both layer and broiler poultry farms, located closer to inland water, had increased odds of experiencing an HPAI outbreak [19]. Iowa also contains a high number of layer and broiler farms, providing further justification to evaluate inland water as a primary risk factor for HPAI in the state [24]. 

As with the Japanese studies, a 2022 South Korean study conducted by Ahmed et al. also evaluated whether there was a significant association between inland water bodies and their distance to poultry farms testing positive for HPAI [20]. Like the Japanese studies, the statistical analysis for this study indicated that proximity to inland water bodies was associated with HPAI outbreaks on poultry farms [20]. The discussion section then echoed an essential theme of the two Japanese studies by reinforcing how waterfowl, attracted to inland water for food and protection, might be shedding HPAI onto susceptible poultry farms as they explore the area for more resources [18,19,20].

Data for these studies were obtained from two Japanese government agencies that granted access to their data: the National Land Numeric Information Download Service and the Japanese Aerospace Exploration Agency [19]. The Korean study conducted by Ahmed et al. had access to outbreak and location data through the “Epidemiology report for the 2016–2017 outbreak of highly pathogenic avian influenza (HPAI) in the Republic of Korea” issued by the Animal and Plant Quarantine Agency (APQA) of South Korea [20]. Data acquisition methods for these studies elucidate how most farm location data is collected by government agencies, but only available to those who work with these agencies. This limits the ability of public health professionals to investigate current HPAI outbreaks and share their results with government agencies and stakeholders.

All three Asian studies elucidated that inland water is a junction where reservoirs and susceptible organisms interact [18,19,20]. These results justify that inland water bodies should be considered a risk factor for HPAI, as inland water bodies are common around poultry farms. They also reinforce how having access to robust location data increases authors’ chances of producing significant results because it allows the authors to incorporate distance into predictor variables [18,19,20].

### 1.4. Canada Geese as a Risk Factor

Canada geese are one of the most abundant species of waterfowl in North America with their ability to thrive in urban, suburban, and rural environments [25]. A recent US study conducted by Hill et al. in 2022, evaluating avian host diversity, reported that Canada geese are worth evaluating as reservoirs and vectors of HPAI because of their high abundance [13]. Hill et al. posited that geese in general are more resilient as hosts, capable of asymptomatic shedding, with the potential to cover longer distances while being infectious [13]. These study results, combined with the fact that the species is one of the most abundant in North America and Iowa, justify evaluating Canada geese as a primary vector for HPAI [4]. The high abundance of poultry farms and inland water in Iowa make evaluation of Canada geese imperative for the state. As the three Asian inland water studies reported, waterfowl will congregate at inland water bodies even if they are close to farms [18,19,20]. Combining those observations with the fact that Canada geese are more gregarious than other species of waterfowl, provides robust justification for their evaluation [25]. The Hill et al. study was funded by the National Institute for Allergies and Infectious Diseases (NIAID) and the authors were able to collect HPAI data in the field via oropharyngeal/nasopharyngeal and rectal/cloacal swabs [13]. Since the authors were in the field, they were able to generate their own location data, allowing for a more robust dataset [13]. While our study was not able to generate location data, we did make attempts to access poultry farm location data.

Iowa also happens to be in a region where waterfowl density is high because of its location along the Mississippi Flyway [4,21,25]. Millions of waterfowl use the flyway to utilize its resources (e.g., inland water bodies, areas to nest) [25]. Because of this, there is a constant intermingling of susceptible and infected waterfowl on the flyway, with Canada geese being one of those species. The Shimizu et al. study on inland water provided evidence for inland water bodies being a location where intermingling of various species resulted in probable transmission and proliferation of HPAI to that specific area [18]. There is extensive documentation on the abundance of Canada geese in Iowa and their comfort around humans and human activity [4,22]. The annual spring population count, conducted by the IDNR, reported Canada geese population of over 80,000 during spring 2023 [21]. It is also common for the species to engage in foraging where human presence and activity is high, like poultry farms [4]. Therefore, as Ahmed et al. concluded in their 2022 study, it is expected that a waterfowl species like Canada geese will be freely wandering around inland bodies close to poultry farms [20].

In 2023 Andrew et al. conducted a study evaluating H5N1 infections in waterfowl from a 2022 outbreak in British Columbia (BC) [26]. Results from this study indicated that Canada geese were the species of wild bird with the highest HPAI detection rate at 32.9% [26]. Their behavior (flocking in large groups) was mentioned as a reason for their high detection rate (Figure 1) [26]. These results, combined with the three Asian studies reporting that waterfowl did flock to inland water close to farms, validate the need to evaluate Canada geese as a primary risk factor. The authors of the BC study also collected HPAI data in the field, via oropharyngeal/nasopharyngeal and rectal/cloacal swabs [26]. Since they were in the field, they generated their location data as well by recording GPS coordinates on the samples [26].

## 2. Materials and Methods

### 2.1. Data Sources

The authors utilized three sources to acquire data for this study. Inland water body surface area data were acquired with assistance from United States Fish and Wildlife Service (US FWS) Iowa field office. Canada geese abundance data were acquired from the Cornell Lab of Ornithology eBird website (eBird). Data on HPAI positive counties were acquired from the Animal Health and Plant Inspection Agency (APHIS). Our initial study design included comparing the locations of inland water bodies and Canada geese abundance, separately, with HPAI positive poultry farm locations. Multiple exposure level variables were to be created for both predictors (e.g., inland water body surface area < 1 km^2^ and >3 km^2^ from a poultry farm).

### 2.2. Period of Evaluation for Iowa: 6 March 2022–20 December 2023

The first reported US domestic HPAI outbreak was in Indiana on 8 February 2022, with the first outbreak in Iowa being reported on 6 March 2022. The US Midwest, where these two states are located, is made up of 12 states and covers approximately 1.9 million km^2^ (APHIS, 2024). Yet, in less than 60 days from the first HPAI positive, the virus was detected in nine (9/12) Midwestern states, including Iowa, with 44% of the domestic HPAI infections being confirmed in Midwestern states [6]. This illustrates how rapidly HPAI can spread to susceptible poultry farms once it is circulating within a geographic region. In Iowa, 94% of all confirmed HPAI poultry farm infections occurred in the state during the reference period of 6 March 2022–20 December 2023 [6]. Only three confirmed poultry farms infections in Iowa have been reported since the reference period. The frequency of HPAI outbreaks during the reference period justifies studying possible environmental risk factors in Iowa. Evaluating these possible risk factors will provide insight into HPAI transmission dynamics for susceptible poultry farms.

### 2.3. Results for Data Acquisition

There were publicly available data for the predictor and outcome variables (inland water surface area, Canada geese species counts, and HPAI outbreaks). Inland water data were available for download at the US FWS wetland mapper website. Canada geese data were available for download on the eBird website. Outcome data were available through the AHPIS public website. eBird and APHIS had websites that were more user-friendly than the US FWS website, and their datasets also required less data cleaning. Downloading data from those sites was also more intuitive than the US FWS website. While county level data were available for HPAI counts, location data for HPAI positive poultry farms were unavailable, and this prevented us from creating variables that incorporated distance.

Without farm location data, this study had to be constructed as an ecological case–control study. We created county level predictor and outcome variables, only using cumulative counts of inland water surface area and Canada geese abundance for all 99 Iowa counties. The outcome variable only identified whether a county experienced an HPAI outbreak. Robust location data for inland water and Canada geese were available, but to maintain consistency among the predictors and outcome of interest, we only evaluated county level comparisons. Furthermore, because we evaluated HPAI outbreaks in Iowa, we used incident density sampling to construct a control group from the same at-risk counties as the cases. Without farm location data, our results on HPAI risk will be imprecise. Access to the addresses or accurate coordinate data of all poultry farms, combined with inland water surface area and Canada geese abundance data, would have allowed for a more robust analysis.

### 2.4. Data Utilization for This Study

#### 2.4.1. Creating Outcome of Interest: HPAI Positive County

The outcome of interest data (HPAI outbreaks) was acquired by visiting the APHIS website. The website contains multiple links to HPAI domestic bird outbreak data, which were embedded on the APHIS HPAI information webpages. The outbreak data contained the name of the state, the name of the county, the month/day/year the outbreak was confirmed, the type of poultry affected, whether the farm was now disease-free, and how many domestic birds were affected (culled). Other than the state and county being identified, there was no location data provided for farms. HPAI outbreak data are still being collected and added to this database. It includes all confirmed outbreaks from February 2022–present. All county level data are fully accessible to the public for viewing and downloading. A dichotomous (1 = yes, 0 = no) HPAI outcome variable was created with this data. The variables represent which counties had an HPAI outbreak during the reference period.

#### 2.4.2. Creating Inland Surface Area Predictor Variable

The first attempt at acquiring inland water surface area data was via the Iowa Department of Natural Resources (IDNR). Their website included county-level data on inland water body surface area dimensions and the type of water body they measured (e.g., river, lake). After examining the data available on the website, the authors realized that the surface area had the exact value for different bodies of inland water that, by definition, were not the same size. A call was made to the IDNR customer service department, and the customer service representative acknowledged that those values were most likely inaccurate, and a review of their data would need to occur. The customer service representative then provided contact information for the US FWS Iowa field office and stated the office could have accurate inland water data for most of the state.

A call was then placed to the US FWS Iowa field office and a researcher from that office provided instructions on how to access inland water surface area data for all 99 counties in Iowa [27]. Wetland water surface data do not capture all inland water bodies in Iowa (e.g., canals, streams), but they do capture a significant amount of inland water surface area data for most of the state. Following the instructions provided, shapefiles containing complete attribute tables for each inland water body considered to be part of US FWS wetlands mapped in the state of Iowa, were downloaded from the US FWS wetland mapper webpage [27]. ARCGIS Pro (ESRI, Berkley, CA, 3.2.2) was used to review the shapefiles. The attribute tables contained inland water sub-type coordinates for the county of origin, region (east or west part of state), and the size of the wetland water body measured in acres. A shapefile from the ARCGIS online portal, that matched all 99 Iowa county names with their coordinates, was used to identify the county of origin for the wetland water bodies listed in the file. The data in the shapefile were divided into counties that occupy the east and west regions in Iowa. The ARCGIS geoprocessing function combined the US FWS inland water surface area data with its corresponding county. The sub-types of wetland water bodies measured were freshwater emergent (water body with vegetation growing above the water surface), freshwater forest shrub (shrubs and young trees growing out of an area that floods), freshwater pond (shallow body of still water), lake (deeper, larger, body of freshwater), and riverine (floodplain that absorbs runoff water, slowly releasing it to streams and rivers) [28]. Continuous and categorical variables can be created with the US FWS data. A histogram was created to evaluate skewness (Figure 3) and it showed the data were slightly skewed. As a result, the data were transformed into a dichotomous variable (1 = yes, 0 = no). The other reason the data were transformed was because the outcome of interest was a dichotomous categorical variable (HPAI positive county), and a chi-square test was selected to test for a possible association. The inland water surface area data we downloaded from the US FWS wetland mapper website was used to calculate the cumulative inland water surface area for each of Iowa 99 counties. We then transformed the data and created the inland water surface area dichotomous categorical variable. Each yes/no response for the dichotomous variable (inland water surface water surface area above mean km^2^) (Table 1) was linked to its corresponding Iowa county (N = 99) by using the HPAI positive county data from APHIS. Data were saved on an SPSS file that was then exported to R Studio 4.3.2. R studio 4.3.2 was also used to create histograms depicting the shape of the US FWS inland water data.

### 2.5. Creating Canada Geese Abundance Predictor Variable

The Cornell Lab of Ornithology maintains a website called eBird that contains general data and information on most bird species in the world. Citizen scientists collect data on birds including species counts of each Iowa county. This website was used to acquire data for the second primary predictor, Canada goose abundance. Data available on the eBird website have been used in published scientific journal articles, indicating that the data are considered of high quality [29,30]. Bird species throughout the world are counted every week, throughout the calendar year, by individuals who register online as spotters for the website [29]. Individuals who register as users are provided with a tutorial on the best places and times to spot birds and how to submit their count observations using eBird checklists. The checklists are available via a cellular phone application (app) that can be downloaded from any of the major online application stores. Once users submit their checklists, the website’s data filter software flags checklists that are abnormal for the time and location submitted. Checklists that are flagged require further documentation (e.g., photos, videos) that is then reviewed by eBird volunteer reviewers, who are birding experts volunteering their time to ensure data collection standards are met [29].

Canada geese count data from the eBird website, for each of Iowa’s 99 counties, were used to create the Canada geese abundance dichotomous variable (yes/no responses for counties with Canada geese counts above the median value) (Table 2). The median value was used to create this variable because the Canada geese data were highly skewed. A histogram was created to visualize and confirm skewness (Figure 4). Clicking on the “explore” tab located on eBird’s home page directs visitors to the data access page where one can choose counts by species, location (county), month, and year(s). Although some Canada geese populations are migratory, we did not incorporate the season into the abundance variable. The reason for this is that Iowa is now home to a substantial population of temperate-breeding Canada geese [31]. There is now a high abundance of these geese in Iowa all year-round, and we evaluated whether this overall abundance was associated with a county experiencing an HPAI outbreak. Data were saved on an SPSS file that was then exported to R Studio 4.3.2 for analysis. R Studio 4.3.2 was used to create histograms depicting the shape of Canada geese data.

### 2.6. Evaluating Data Cleaning

An article written by Van den Broeck et al. in 2005 provided a framework for evaluating data cleaning [32]. We utilized this framework to evaluate how labor-intensive data cleaning was for the outcome of interest (HPAI outbreaks) and the predictor variables (wetland water surface area, Canada geese count). Our evaluation focused on the analysis of secondary datasets. A 5-point Likert scale was utilized to rate the aspect of data cleaning (1 = very labor-intensive data cleaning, 2 = labor intensive for data cleaning, 3 = data cleaning was not labor intensive, nor was it simple, 4 = simple data cleaning needed, to 5 = almost no data cleaning needed).

## 3. Results

### 3.1. Data Evaluation Results

There were publicly available data for the predictor and outcome variables (inland water surface area, Canada geese species counts, and HPAI outbreaks). Inland water data were available for download at the US FWS wetland mapper website. Canada geese data were available for download on the eBird website. Outcome data were available through the AHPIS public website. eBird and APHIS had websites that were more user-friendly than the US FWS website, and their datasets also required less data cleaning. Downloading data from those sites was also more intuitive than the US FWS website. While county level data were available for HPAI counts, location data for HPAI positive poultry farms were unavailable, and this prevented robust evaluation of both predictor variables.

The highest data cleaning score was for the HPAI outcome data collected from APHIS website (11/15). The second highest score belonged to the dataset collected from the eBird website (8/15), and the lowest score belonged to the dataset collected from the US FWS website (6/15). Lower scores corresponded with the number of steps it took to access data for each variable (Table 3, Table 4 and Table 5). The lowest data cleaning score (6/15) belonged to the US FWS data, and data for this variable also had the greatest number of steps to access the dataset. Table 6, Table 7 and Table 8 provide a detailed explanation of the steps taken to access and acquire data for each variable.

### 3.2. Predictor Results

The Pearson’s chi-square test evaluating whether there was an association between Canada geese counts above the median and HPAI outbreaks (Table 9), was significant (X^2^ = 4.29, *p* = 0.04). However, the contingency table for this result indicated that HPAI negative counties were associated with Canada geese counts above the median count (expected counts below median = 14, actual count = 19). The post chi-square test One Health diagram (Figure 5) displays a visual summary of our study’s unexpected results. No significant association was present for inland water surface area sum (km^2^) above the mean and HPAI outbreaks.

## 4. Discussion

### 4.1. Data Acquisition

We were able to locate and access publicly available data for the predictor and outcome variables (inland water surface area, Canada geese species counts, and HPAI outbreaks). Downloading Canada geese abundance data from the eBird website and HPAI outcome data from the APHIS website was more user-friendly than downloading inland water data from the US FWS website. While county level data were available for HPAI outbreaks, location data for HPAI positive poultry farms were unavailable. This prevented robust evaluation of both predictor variables. The lack of location data is an issue that needs to be addressed. Not having location data hinders comprehensive research into the possible levels of risk that might be associated with poultry farm proximity to inland water, or to a high abundance of Canada geese.

### 4.2. Attempts to Access Location Data

Since APHIS was involved in surveillance, collection and HPAI outbreak data entry, the authors reached out to APHIS and their parent agency, the United States Department of Agriculture (USDA). We inquired about obtaining poultry farm location data in the form of addresses or accurate GPS coordinates, to be used with the primary predictors and outcome of interest. The authors attempted to obtain infected farm address data by contacting the USDA’s customer service email portal, the customer service call center, and the APHIS Office of Public Affairs. The response from all sources was that APHIS would not release the location of infected farms because of confidentiality laws.

The authors then attempted to use the World Organization of Animal Health’s website for location data, since a link to the website was also embedded on the APHIS website. WOAH was working with APHIS on conducting HPAI surveillance, and their website did contain location data in the form of coordinates that represented where an outbreak farm was supposed to be located. These coordinates were not accurate, however, as when they were entered into Google Maps the images of the surrounding area showed no farms present at those locations. The authors then attempted to clarify whether the coordinates were accurate, by contacting the WOAH delegate for the Americas. A response was never received; therefore, the seemingly inaccurate location data could not be used.

Since a comprehensive evaluation of the study’s primary predictors was not possible without location data, we then attempted to access that data by drafting a Freedom of Information Act (FOIA) letter (Appendix A). The letter was sent to APHIS parent agency the USDA. Instructions on the USDA website stated that the letter needed to be sent to the agency’s Office of Freedom of Information & Privacy Act, Legislative and Public Affairs. The FOIA letter sent to this department requested de-identified location data for all poultry farms that experienced an HPAI outbreak in Iowa. A response was received describing that APHIS would provide the same data found on their website dashboard with two additional data columns: how a farm culled infected/exposed birds, and the amount of indemnity paid (Appendix A). No explanation was given as to why address data were not included. A call was placed to the USDA’s Iowa office to ask why location data would not be provided. The customer service staff member referred the authors to Iowa state law 163.3C (Appendix A) [33].

Iowa state law 163.3C allows government officials, and those representing state agencies, to deny access to information on foreign animal diseases. Since H5N1 originates from Asia it is classified as a foreign animal disease [33]. The reason given for this law is that sharing sensitive information, such as the location of farms that experienced outbreaks, could hinder the state’s response efforts.

Without access to farm location data, robust geospatial analysis could not be carried out. This hindered our ability to provide important information to stakeholders as the three Asian inland water studies had done for stakeholders in their respective regions [18,19,20]. Those studies were able to provide distances at which the odds of an HPAI outbreak increased, informing their stakeholders where intervention efforts should be focused [18,19,20]. The lack of location data also hindered our ability to evaluate Canada geese abundance, as we could not test for the significance between where the HPAI positive farms were located and where the Canada geese counts occurred. Not having location data is relevant because, as Luukkonen et al. reported in their 2022 study, Iowa is now home to a substantial population of temperate-breeding Canada geese [31]. Understanding how proximity of those populations to susceptible poultry farms can affect HPAI risk will provide stakeholders with the information they need to make informed decisions on which prevention methods to utilize and where to utilize them.

Incorporating farm location data with inland water and Canada geese data would have provided the opportunity to create detailed variables that would have allowed us to perform a more comprehensive evaluation. Even if the results were statistically insignificant, there is practical significance to reporting any results from a comprehensive evaluation. The results will help stakeholders determine if their farms are at greater or lesser risk based on their proximity to inland water or large populations of Canada geese. Future evaluation of these factors using location data should be prioritized until there is statistical evidence that both these environmental factors are not significant.

### 4.3. Data Cleaning Evaluation

We utilized the framework from the Van den Broeck et al. 2005 study on evaluating data cleaning methods. We scored our data cleaning efforts based on this framework because it provided criteria for identifying common data management issues, such as the need for analysis deletions, excess or lack of data, and data transfer errors [32]. The results of the data cleaning evaluation revealed that more steps to access data also resulted in lower scores. Lower scores indicated more data cleaning was needed. APHIS data required the least amount of cleaning, and eBird data the second least amount of data cleaning. US FWS data required the most amount of data cleaning and required the greatest number of steps to access. A possible reason that less data cleaning was needed for the APHIS data is because, as a federal agency tasked with monitoring HPAI outbreaks in real time, they have more resources allocated to their data management. The amount of data cleaning needed for US FWS data is an issue that warrants further evaluation. Data downloads from the US FWS should be user-friendly and require minimal cleaning since they are a federal agency tasked with monitoring the health of important environmental domains, such as inland water and habitat protection. Reduced data cleaning can result in prompt evaluation of pertinent HPAI research where timely results are needed.

The US FWS inland water data required reformatting to be recognized by R and knowledge of ARCGIS Pro was also needed. This was more labor intensive compared to the process of accessing, downloading, and using the APHIS and eBird data. After downloading the inland water body data from the US FWS Wetland Inventory Mapper website, ARCGIS had to be used to extract data from the shapefile format. Once extracted, the data needed to be labeled after the counties in Iowa that they were representing. The file to combine the labels with the data needed to be searched for within the ARCGIS online portal. After locating the labels, ARCGIS spatial joining tools needed to be applied. The dataset formed from merged tables was viable after using the ARCGIS tools to merge the data. While this process was more time-consuming than the processes for accessing outbreak and species count data, the resulting product was of high quality. All inland water bodies mapped by the US FWS, for all 99 Iowa counties, were represented in the dataset. Surface area and latitude/longitude coordinates for each water body were also part of the dataset. Despite minor challenges, accessing data for HPAI outbreaks, inland water body surface area, and Canada geese species count, were successful endeavors.

### 4.4. Evaluation of Inland Water and Canada Geese Abundance

Producing significant Pearson’s chi-square test results also validated the data acquisition methods used in this study. Despite data limitations, we were able to provide evidence of an association between Canada geese abundance and HPAI negative counties. The results were unexpected, as Canada geese counts above the median value were associated with HPAI negative counties, indicating their presence might be protective. However, the significant results did justify identifying our predictor as worth further evaluation, which aligns with the conclusions from the Hill et al. study [13]. Our study and the Hill et al. study elucidate how much is unknown about a common waterfowl species that might affect HPAI transmission dynamics [13]. Both studies provide evidence for further evaluation of Canada geese to provide a better understanding of whether this species is associated with higher or lower risk of an HPAI outbreak. Our results indicated that they may be associated with lower risk, but the best way to evaluate risk is to do so by incorporating the proximity of the Canada geese to the HPAI positive farms.

Another reason higher counts of Canada geese may not indicate increased risk of HPAI outbreaks is because larger groups of Canada geese may congregate in open areas away from farms, whereas smaller HPAI infected flocks may visit areas that are closer to the farms. As reported in the 2022 Luukkonen et al. study, Canada geese do seek out agricultural fields, so abundance might not indicate risk, but their presence at certain distances around the perimeter of poultry farms might be associated with higher HPAI risk [31]. The only way to rule out that possibility is to conduct a more thorough evaluation with farm location data. It needs to be incorporated into the datasets described in this study, so that geospatial analysis between Canada geese and HPAI positive poultry farms can be conducted. Furthermore, if we had access to location data it is possible that incorporating that data with the inland water variable (e.g., wetland water surface area above the median value and within 1 km^2^ of a poultry farm) could have produced significant results.

### 4.5. Primary Study Limitation: Inability to Access Location Data

The primary limitation to a comprehensive evaluation of the variables of interest was the lack of poultry farm location data for the outcome variable (HPAI outbreaks). Having the location data for HPAI positive and negative farms would have enabled a more robust evaluation of risk for the primary predictors because it would add geospatial context to these variables. Since our study did not have access to location data for poultry farms, we could only conduct a county level evaluation of inland water surface area in Iowa. This limited the ability of our study to produce significant results. Only results that incorporate robust location data, such as those from the three Asian inland water studies, can benefit stakeholders and public health.

Having farm location data allowed all three Asian inland water studies to provide detailed results on whether an association existed between HPAI risk and the distance in meters from inland water bodies to poultry farms [18,19,20]. Being able to identify if risk changes by distance to inland water is the type of detail that lets stakeholders know where to implement prevention measures (e.g., where to place barriers, where to drain smaller water bodies). Without location data, this study was only able to evaluate whether there was a general association between the predictor variables and the HPAI outbreaks.

Having access to location data would have also improved evaluation of the Canada geese variable. Although this variable yielded significant results, having location data would have allowed us to identify the distance at which the species’ abundance affects risk. Instead of creating a categorical variable based on median values alone, we could have created variables based on the combination of Canada geese counts (>100), and their proximity to poultry farms (counts < 100, 50 m from farms). This would have allowed for a similar evaluation as those found in the Asian inland water studies, where risk was measured for different distances [18,19,20]. It is vital to evaluate Canada geese counts and proximity together because the species is terrestrial and comfortable with human activity. Their presence might not increase risk from afar, but that could change if they are closer to a farm.

### 4.6. Limitations Specific to Predictor

The unique limitation of the US FWS inland water data was that the data were obtained from analysis conducted at high altitudes using aircraft surveillance. Wetlands are identified by aircraft based on vegetation, visible hydrology, and geography. On-the-ground inspections are employed to confirm ariel data, but this was not done for all water bodies. Other limitations include wetland features that may have changed since the date of data collection. Furthermore, some inland water habitats are excluded from the national mapping program because of the limitations of aerial imagery [27]. When the initial aerial photos are taken, inland water body details (depth, whether vegetation is growing under water surface) cannot be confirmed until ground observations are conducted. As a result, their dimensions will not be added to the US FWS shapefile until the ground observation occurs.

The Canada geese abundance data had a unique limitation in that eBird data are recorded using observations collected from citizen-scientist. Anyone who takes an interest in bird watching and counting can submit an eBird checklist; expertise in ornithology is not necessary. The lack of expertise involved in collecting data is offset by eBird employing “expert data reviewers,” but these reviewers still rely on individuals to learn and follow the reporting protocols posted on the website [29]. They also do not have control over how people will process and perceive the protocols, which could lead to errors in reporting. Even with these limitations, eBird data have been used in other scientific studies [29]. Given the dearth of publicly available robust bird data, it is still the foremost choice for publicly available waterfowl data.

### 4.7. Possible Solution for Granting Public Access to Location Data

The methods reviewed in this study yielded usable data that provided the opportunity to begin an evaluation of inland water bodies, Canada geese abundance, and whether these factors were associated with HPAI outbreaks in Iowa. However, future studies conducted by qualified researchers not employed by government agencies need access to the locations of HPAI negative and positive poultry farms. Allowing public access to qualified researchers, while still protecting the privacy of farm owners and upholding the primary function of Iowa state law 163.3C, can be done. The template for how to achieve this can be found in how the US protects personal medical information using the Health Insurance Portability and Accountability Act (HIPAA). To access HIPAA protected information, healthcare workers need to be trained on how to handle the information, what the restrictions for use are, attest that they understand how to use the information, and will accept responsibility for the legal consequences they will face if they misuse the information. An analogous system can be created for accessing sensitive agricultural information, like the location of HPAI positive and negative poultry farms. The process can take place online, where researchers requesting the data will need to create an account, complete training that reinforces how to use the data, and attest that if the data are misused, they understand the legal consequences they will face.

Creating a pathway for public access to accurate farm location data will allow for more comprehensive evaluations of natural environmental factors unique to agriculture. Having a better understanding of these factors will help public health professionals and stakeholders develop, or reinforce, interventions to decrease HPAI incidence among farm workers, domestic birds, and the public.

## 5. Conclusions

The expected results for inland water and Canada geese abundance variables did not materialize. However, the significant results indicating a protective association between Canada geese abundance and HPAI outbreaks did elucidate the need to acquire accurate location data. Acquiring accurate location data will provide subsequent authors with the ability to create and test more detailed variables. Creating a pathway for public access to accurate farm location data will allow comprehensive evaluations of natural environmental factors unique to HPAI outbreaks. Having a better understanding of these factors will help public health professionals, and stakeholders, decrease HPAI incidence among farm workers and domestic poultry, as well as reduce the transmission risk among the public.

## Figures and Tables

**Figure 1 ijerph-22-00400-f001:**
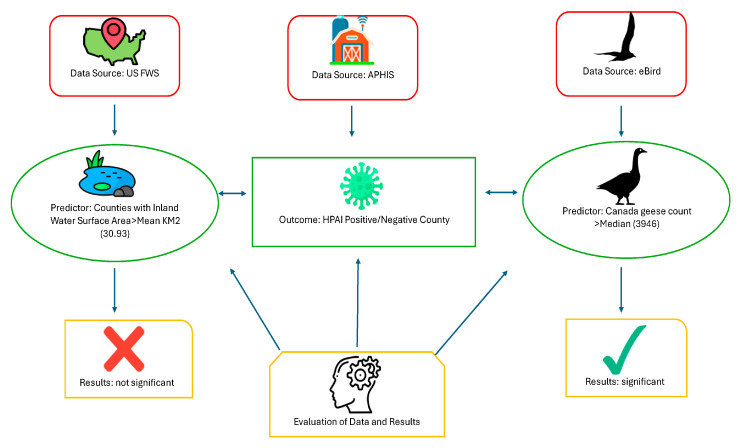
Semantic diagram of study.

**Figure 2 ijerph-22-00400-f002:**
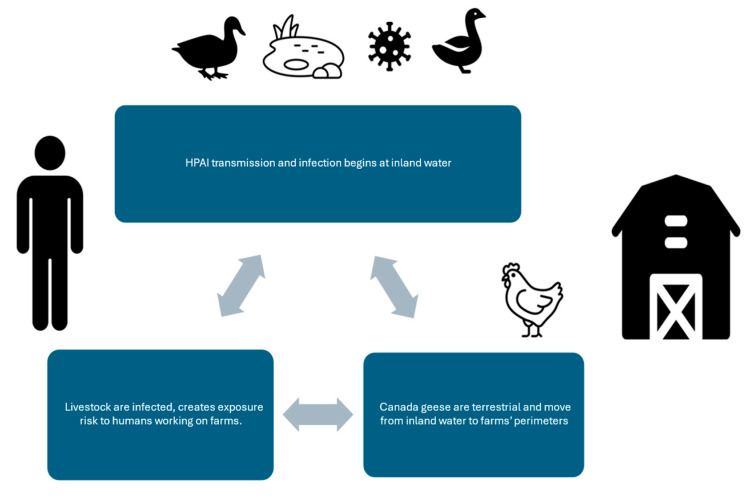
One Health diagram of study variables and possible outcomes.

**Figure 3 ijerph-22-00400-f003:**
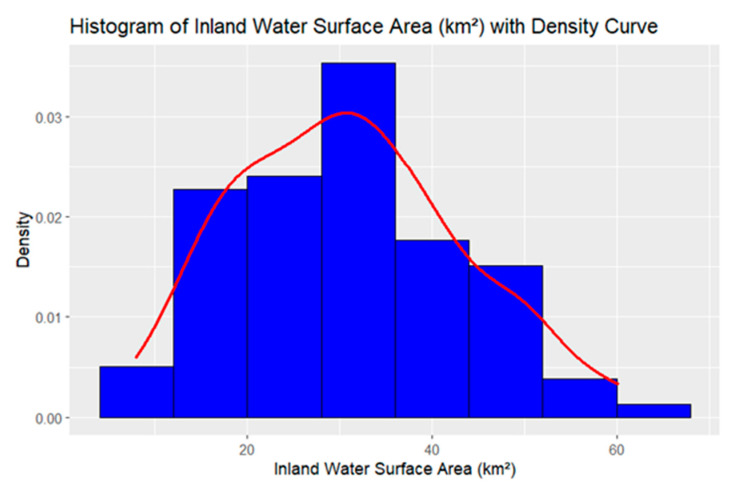
Histogram for inland water data.

**Figure 4 ijerph-22-00400-f004:**
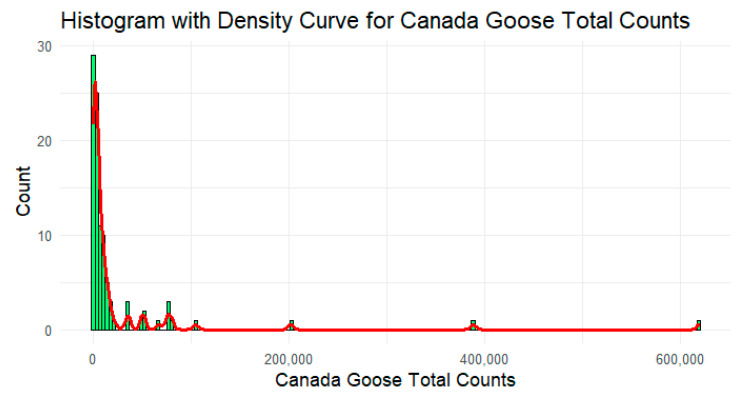
Histogram for total Canada geese counts during reference period.

**Figure 5 ijerph-22-00400-f005:**
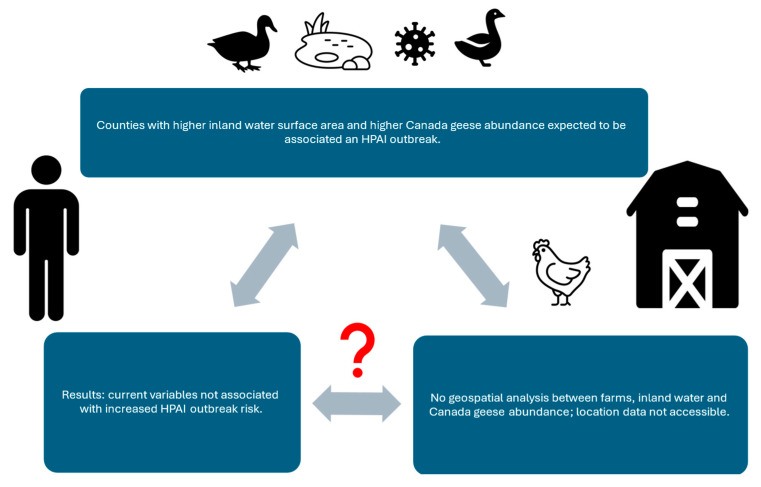
One Health diagram of study variables results.

**Table 1 ijerph-22-00400-t001:** Inland water surface area median for HPAI positive and negative Iowa counties.

Variable	*N* (Counties)	Mean km^2^ (SD)
Inland water body surface Area median HPAI positive	28	28.12 (12.57)
Inland water body surface Area median HPAI negative	71	32.04 (11.73)
Inland water surface area Median all counties	99	30.93 (12.07)

Counties not listed in the APHIS data as HPAI positive were considered HPAI negative counties. Median km^2^ for US FWS inland water data was calculated with R Studio 4.3.2.

**Table 2 ijerph-22-00400-t002:** Canada geese counts above median value for HPAI positive and negative Iowa counties.

Variable	*N* (Counties)	Median km^2^ (SD)
Canada geese count median HPAI positive	28	2607 (21,152)
Canada geese count median HPAI negative	71	5356 (89,228)
Canada geese median all counties	99	3947 (76,673)

**Table 3 ijerph-22-00400-t003:** Data cleaning evaluation for APHIS website data.

APHIS Website	Lack of, or, Excess Data Carried over.	Data Extraction or Transfer Error.	Deletions or Duplications by Analysis.
Analysis score	2	5	4
Total score	11		

**Table 4 ijerph-22-00400-t004:** Data cleaning evaluation for eBird website data.

eBird Website	Lack of, or, Excess Data Carried over.	Data Extraction or Transfer Error.	Deletions or Duplications by Analysis.
Analysis score	2	4	2
Total score	8		

**Table 5 ijerph-22-00400-t005:** Data cleaning evaluation for US FWS website data.

US FWS Website	Lack of, or, Excess Data Carried over.	Data Extraction or Transfer Error.	Deletions or Duplications by Analysis.
Analysis score	1	1	4
Total score	6		

**Table 6 ijerph-22-00400-t006:** Steps to access HPAI outcome data from APHIS.

APHIS HPAI Outcome Steps to Access Data
1. Search “HPAI outbreaks”, on APHIS home page, click on avian influenza in search results
2. Find and enter “current status” tab after being redirected
3. Enter “Confirmations in Commercial Flocks” tab. Scroll down to data

**Table 7 ijerph-22-00400-t007:** Steps to access Canada geese abundance data from eBird.

eBird Predictor Steps to Access Data
1. Enter “Explore” tab on eBird home page
2. Click and enter “Bar Charts
3. Select state of interest
4. Select county of interest
5. Choose “Line Graph” data
6. Choose “Download Line Graph” data

**Table 8 ijerph-22-00400-t008:** Steps to access inland water data from US FWS.

Inland Water Predictor Steps to Access Data
1. Search “Wetland Inventory” on US FWS home page, click on wetland inventory in search results
2. Click and enter “Get Data” tab
3. Select state of interest
4. Select shape file for wetland waterbodies for selected state
5. Download and save shapefile
6. Need ARCGIS Pro
7. Open ARCGIS pro, select map/new project and open all relevant wet-land water body shapefiles
8. Shapefile of wetland water bodies is represented on map as shade area, search ARGGIS online portal for counties within state of interest
9. Add ARCGIS shapefile with county boundaries as layer on map
10. Select “Analysis” and “Tools”. Search for spatial join tool on search bar provided. Select “Spatial Join” tool
11. In “Spatial Join” tool dialogue box, select wetland water bodies as the “Target Feature” and count boundaries as the “Join Feature”

**Table 9 ijerph-22-00400-t009:** Pearson’s chi-square results, testing for association between all Iowa counties (N = 99) with Canada geese counts above the median and inland water surface above the mean, and whether they were also HPAI positive.

Variable	X^2^	*p* Value
Counties with Canada geese count > median	4.29 *	0.04 *
Counties with inland water km^2^ > mean	0.14	0.71

* Significant, *p* < 0.05.

## Data Availability

Direct links are provided to website pages where raw data for variables of interest are being stored. SPSS datasets imported to R Studio 4.3.2 are included in the folder with the Appendix A (R code.docx and Rdata.sav). HPAI outbreaks: https://www.aphis.usda.gov/livestock-poultry-disease/avian/avian-influenza/hpai-detections/commercial-backyard-flocks (accessed on 2 January 2024). Inland water surface area: https://fwsprimary.wim.usgs.gov/wetlands/apps/wetlands-mapper/ (accessed on 31 January 2024). Canada Geese count: https://ebird.org/barchart?r=US-IA&bmo=1&emo=12&byr=2022&eyr=2023&spp=cangoo (accessed on 2 April 2024).

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
