# Peer review of "Exploring Methods to Evaluate HPAI Transmission Risk in Iowa During Peak HPAI Incidence, February 2022–December 2023"

_ijerph, 2025, doi:10.3390/ijerph22030400_

Round 1
Reviewer 1 Report
Comments and Suggestions for Authors
– The topic of the manuscript is of interest. It examines the environmental risk factors contributing to the spread of H5N1 HPAI in Iowa, the state most affected during the 2022–2023 epizootic. It aims an approach to assess the significance and data availability of inland water surface area and Canada geese abundance during the peak incidence period.
– The topic is original and relevant to the field. This study focused on methods to evaluate a significant association between inland water bodies, Canada geese abundance, and the risk of HPAI during the HPAI. This study specifically aimed to evaluate these two non-human factors, as current evidence suggests that human activity is not playing a substantial role in HPAI transmission, unlike during the previous 2014–2015 epidemic. The significant results from the Chi-Square test for the Canada geese predictor also provide evidence that non-human factors need further investigation.
– This study provides a significant evaluation on non-human factors, specifically inland water bodies and Canada geese abundance, in the context of HPAI transmission during the HPAI-21. Unlike previous studies, which primarily emphasized human-driven factors during earlier epidemics such as the 2014–2015 outbreak, this research highlights the potential significance of HPAI transmission. The findings, supported by strong statistical evidence to understand non-human contributors to HPAI.
– Yes, the conclusions are consistent. The study effectively links the observed association between inland water bodies, Canada geese abundance, and HPAI risk to the need for further evaluation of non-human factors. However, the conclusion is large and can be improved according to the journal’s format.
– References are appropriate.
– Overall, the manuscript is well-structured, and the findings are supported by robust data analysis techniques.
Minor revision:
1. A semantic diagram would be beneficial to enhance the understanding of the study design detailed in lines 111 to 122 on page 3.
2. Table 7 could be improved by presenting the data in a more organized and well-arranged format.
3. The discussion section lacks sufficient support citations; it should be strengthened by incorporating references to relevant studies reported earlier.
Author Response
Comment: A semantic diagram would be beneficial to enhance the understanding of the study design detailed in lines 111 to 122 on page 3.
Response: Semantic diagram is now included in the manuscript on line 53.
Comment: Table 7 could be improved by presenting the data in a more organized and well-arranged format.
Response: Table 7 has been changed to three separate tables starting at line 450.
Comment: The discussion section lacks sufficient support citations; it should be strengthened by incorporating references to relevant studies reported earlier.
Response: Discussion section now includes support citations starting on line 519.
Reviewer 2 Report
Comments and Suggestions for Authors
I read with great interest the proposed article titled: "Exploring Methods to Evaluate HPAI Transmission Risk in Iowa During Peak HPAI Incidence, February 2022–December 2023" by Christopher Jimenez and Lori A. Hoepner.
Overall:
Since HPAI is recognized as having a high risk for a pandemic in the future, this study highlights the need for investment in HPAI monitoring. The article is well-written and well-presented but I think the parts such as the Introduction and the conclusion could be shorter than they are. The authors can only present concise information in both the introduction and the conclusion parts.
Details:
Lines 9-14: I would like to ask the authors to give a summary of their results here but also one sentence summarizing their conclusion.
Lines 84-109: It is hard to discern the specific problem addressed by the author. Instead of focusing on a clear research question, they have presented various results from other studies without synthesizing or aligning them toward a specific objective. Please, clarify here the "research question" and the study's aim.
Lines 127-132: It seems that the authors presented the results in this part. I will suggest moving this part to the result section.
Lines 193-218: I am concerned about the methods the authors used for the calculation of the abundance of Canada geese. Indeed, Canada geese are migratory animals and their migration behavior varies depending on the population and location. I would like to know if the authors also considered the "time factor" or seasonality in their estimation.
Author Response
Comment: Lines 9-14: I would like to ask the authors to give a summary of their results here but also one sentence summarizing their conclusion.
Response: Abstract now includes results summary and conclusions.
Comment: Lines 84-109: It is hard to discern the specific problem addressed by the author. Instead of focusing on a clear research question, they have presented various results from other studies without synthesizing or aligning them toward a specific objective. Please, clarify here the "research question" and the study's aim.
Response: synthesis of articles used and a clearer description of study objectives have now been included. Synthesis now starts at line 149, study objectives discussed on line 32.
Comment: Lines 127-132: It seems that the authors presented the results in this part. I will suggest moving this part to the result section.
Response: All results related content has been moved to the results section starting at line 394.
Comment: Lines 193-218: I am concerned about the methods the authors used for the calculation of the abundance of Canada geese. Indeed, Canada geese are migratory animals and their migration behavior varies depending on the population and location. I would like to know if the authors also considered the "time factor" or seasonality in their estimation.
Response: Clarification on why season/time factor was not included in this study explained on lines 365-376.
Reviewer 3 Report
Comments and Suggestions for Authors
Exploring Methods to Evaluate HPAI Transmission Risk in 2 Iowa During Peak HPAI Incidence, February 2022–December 3 2023
The current manuscript evaluates methods for assessing the transmission risk of High Pathogenic Avian Influenza (HPAI) in Iowa during its peak incidence (February 2022 – December 2023). The study focuses on two key environmental factors—Canada geese abundance and inland water surface area—as potential contributors to HPAI outbreaks among domestic poultry. This study underscores the need for further investigation into non-human factors influencing HPAI transmission. While the research establishes a connection between Canada geese presence and outbreaks, it calls for improved data accessibility and more detailed spatial analysis to refine risk assessments and enhance outbreak preparedness.
However, here are some suggestions to improve the manuscript:
Line # |
Comments |
Abstract |
It may be improved. Currently, it lacks any results and conclusion parts. |
|
|
38-39 |
“However, human-to-human transmission might eventually occur as all strains of HPAI are consistently mutating”
The following references may serve to emphasize the occurrence of mutation due irrational use of antiviral drugs.
Neuraminidase Inhibitors Resistance: The Irrational Use of Oseltamivir Can Lead to Genesis of Mutant Avian Influenza Viruses in The Field
Avian Influenza in Low and Middle-Income Countries (LMICs): Outbreaks, Vaccination Challenges and Economic Impact
|
76-83 |
References from the mentioned 3 studies may be incorporated here. |
109 |
Introduction may end up with a paragraph summarizing the objectives of the study. |
162 |
a researcher for that office
a researcher from that office |
Discussion |
Discussion lacks any comparison with previous studies. This part may be improved by comparing the results of the current studies with already recently published studies. |
Conclusions |
This part me reduced to a paragraph with brief summarizing the conclusions. Currently, there are more than 6-7 paragraph of the conclusions. |

Author Response
Comment: Abstract It may be improved. Currently, it lacks any results and conclusion parts.
Response: Abstract now includes results summary and conclusions.
Comment: “However, human-to-human transmission might eventually occur as all strains of HPAI are consistently mutating” The following references may serve to emphasize the occurrence of mutation due irrational use of antiviral drugs.
Response: articles have been incorporated into the introduction section starting at line 115.
Comment: References from the mentioned 3 studies may be incorporated here.
Response: References incorporated on lines where this paragraph now resides, line 182.
Comment: Introduction may end up with a paragraph summarizing the objectives of the study.
Response: study objectives now summarized on line 32.
Comment: a researcher for that office- suggested change: a researcher from that office
Response: change made.
Comment: Discussion lacks any comparison with previous studies. This part may be improved by comparing the results of the current studies with already recently published studies.
Response: Discussion now incorporates previous studies starting on line 519.
Comment: Conclusions This part me reduced to a paragraph with brief summarizing the conclusions. Currently, there are more than 6-7 paragraph of the conclusions.
Response: Conclusion section now includes only 7 lines starting at line 680.